# A Novel Sub-Pixel-Shift-Based High-Resolution X-ray Flat Panel Detector

**Jiayin Liu** 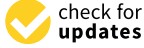 **and Jae Ho Kim** *

Image and A. I. Laboratory, Department of Electronics Engineering, Pusan National University, Busan 46241, Korea; liujiayinpnu@gmail.com
* Correspondence: jhkim@pusan.ac.kr; Tel.: +82-10-4042-2450

**Abstract:** In this paper, we describe a novel sub-pixel shift (SPS)-based X-ray flat panel detector (FPD), which can achieve high resolution while maintaining a high SNR (signal-to-noise ratio). In the proposed architecture, an XY precision shift stage is applied to complete the sub-pixel shift process. In addition, image acquisition and high-resolution image composition are integrated in the FPD hardware. According to the relevant standards for detector image quality evaluation, we tested and evaluated some image quality indicators. The results show that the proposed FPD with SPS outperforms the original FPD without SPS technology. More specifically, the measured pixel size of the proposed FPD was reduced from 162 to 140 μm for $2 \times 2$ sub-pixel shift mode, and 132 μm for $4 \times 4$ sub-pixel shift mode, that is, the basic spatial detector resolution was improved by 13.6% for the simplest $2 \times 2$ sub-pixel shift mode, and by 18.5% for $4 \times 4$ sub-pixel shift mode. With this method, a lower-price FPD is elevated both in resolution and $SNR_n$ to meet imaging quality requirements.

**Keywords:** X-ray flat panel detector; high resolution; sub-pixel shift

## 1. Introduction

The X-ray flat panel detector (FPD) [1,2] is a photographic element used in digital radiography. In the same way that a normal digital camera uses a CMOS sensor to receive light passing through a lens and converts it into an image, a flat panel detector converts X-rays passing through the object into a digital image.

X-ray flat panel detectors have been widely used in security, industrial, and medical applications in place of conventional image intensifiers (I.I.) [2] and imaging plates (IP) [3]. The dynamic range of FPD is greater than that of I.I., and the images can be viewed in real time, without the need to remove the plate and extract the images, as with IP. The main applications of FPD in the industrial field are X-ray and CT non-destructive inspection, including casting and welding inspection, 3D printing inspection, SMT (surface-mounted technology) and semiconductor inspection, new energy battery inspection, security check, and so on; in the medical field, the main applications cover almost all X-ray equipment, including DR (digital radiography), DRF (dynamic DR), DM (digital mammography), CBCT (dental CT), DSA (digital subtraction angiography), C-arm X-ray systems, and so on.

The first flat-panel detector DR systems based on amorphous silicon [4] and amorphous selenium [5] were introduced in 1995. Subsequently, major medical imaging equipment companies conducted preliminary research on the technology. In the late 1990s, GE and Perkin Elmer in cooperation, Thales, Siemens, Philips in co-investment with Trixell, Varex, Canon Medical, and other companies developed amorphous silicon flat panel detectors [6]. Around 2010, amorphous silicon flat panel detector technology further proliferated, and traditional film giants Carestream, Fujifilm, Konica, and Agfa also developed flat panel detectors. Meanwhile, South Korea's Viewworks and Rayence and China's PZImaging, KangZhong, and i-Ray also launched their own amorphous silicon flat panel detectors [7].

Several kinds of FPDs have been developed over the past few decades [8]. Existing FPDs are divided into two types: indirect conversion detectors and direct conversion detectors [9]. The principle of indirect conversion FPDs is that X-ray irradiation is first converted to visible light through the scintillator; then, the digital image is read out using the principle of visible light cameras. Its basic structure includes the following: scintillator, sensor and readout circuit, and peripheral control circuit. The scintillator and sensor are the core part and determine the main performance of the FPD. Amorphous silicon (a-Si), CMOS, IGZO [10], and flexible FPD are all indirect conversion detectors. In contrast, direct conversion FPDs do not require scintillators. They convert X-rays into electrical signals directly after the X-rays are collected by the photoconductive semiconductor material. Therefore, the basic structure of the direct conversion FPD includes the following: sensor and readout circuit, peripheral control circuit, and the sensor (photoconductive semiconductor), which is the core part.

The technologies used by different types of detectors and their main application areas are shown in Table 1.

**Table 1.** The technologies used by different types of detectors and their main application areas.

| Major Categories | Types | Detector Technology | Main Application Areas |
|---|---|---|---|
| Indirect conversion detectors | a-Si FPD | Scintillator + a-Si + TFT | DR/DRF, radiotherapy, industrial |
| | IGZO FPD | Scintillator + a-Si + IGZO | DR/DRF, intervention |
| | CMOS FPD | Scintillator + CMOS | Dental, mammograph, surgical, industrial |
| | flexible FPD | Scintillator + a-Si + TFT | Mobile healthcare |
| Direct conversion detectors | a-Se FPD | a-Se + TFT | Mammograph |
| | photon counting FPD | CdTe/CZT +CMOS | CT, breast CT |

The two most important metrics to evaluate the imaging quality of FPDs are the signal-to-noise ratio (SNR) [11] and spatial resolution (SR) [12]. FPDs have different SNR and SR due to different materials, structures, and processes. SNR affects the ability to distinguish density differences in different tissues (i.e., high SNR means high density resolution), and SR affects the ability to distinguish fine spatial structures of tissues. To improve the spatial resolution, the pixel size of the detector needs to be made smaller. However, too small a pixel size leads to a decrease in SNR. As a direct consequence, the density resolution decreases, the image signal-to-noise ratio deteriorates, and the image becomes unusable. It is generally necessary to find a balance between these two imaging indexes. There is no FPD on the market that performs well in both SNR and spatial resolution.

In the medical field, the target of DR equipment is to observe and distinguish the density of different tissues in the chest and lungs. Therefore, the requirements for density resolution (SNR) are relatively high, while the pixel size of FPDs are larger, generally 139 μm, to easily obtain images with higher contrast. Accordingly, amorphous silicon FPDs are generally selected, which are more conducive to diagnosis. For the examination of extremity joints and breast, better imaging of structural spatial details is needed, so the pixel size of the FPD should be small, generally 50 to 76 μm, in order to obtain high-spatial-resolution images. Generally, amorphous selenium or CMOS FPDs are selected for these types of applications. The main reason for the higher requirement of spatial resolution in the industrial field than in the medical field is the continuous improvement of product quality control requirements. Defects that need to be detected are getting smaller and smaller as the result of the process upgrading. At present, film imaging is still used in a large number of industrial inspection scenarios, mostly on weld and casting inspection. In addition to regulatory factors, the main reason why users choose film over FDPs is that the actual spatial resolution of FPDs cannot reach the film imaging level. The industry expects that the spatial resolution could be further improved without decreasing SNR.

The resolution of the original image can be improved by pure software methods, which some people call super-resolution. Currently, there are three methods to achieve image super-resolution by software: interpolation-based methods, reconstruction-based methods, and learning-based methods.

Interpolation-based methods try to increase image resolution by filling in the corresponding pixel values on the empty spots after zooming in. However, these methods are efficient but ineffective [13,14]. Reconstruction-based methods align multiple low-resolution images of the same scene with sub-pixel accuracy on the space, obtaining the motion offsets between high- and low-resolution images and constructing the spatial motion parameters in the observation model to obtain a high-resolution image [15–19]. The core idea of reconstruction-based super-resolution methods is to trade temporal bandwidth (acquiring multiple image sequences of the same scene) for spatial resolution. With the significant development of deep learning technology, image super-resolution technology does have a wide range of practical applications in the fields of games, movies, medical imaging, and so on. Learning-based methods adopt the end-to-end mapping function of low-resolution images to high-resolution images by neural networks. Using the prior knowledge acquired by the model to obtain high-frequency details of the image, they are considered the best way to enhance image resolution at present. The mainstream algorithms are SRCNN, SRGAN, ESRGAN, and so on. Among them, the SRCNN method [20,21] has the simplest network structure, using only three convolutional layers, and the framework is flexible in choosing parameters and supports customization. The disadvantage is that the details are not sufficiently represented, and the results obtained are too smooth and unrealistic when the magnification exceeds four. The SRGAN method [22] generates realistic textures for a single image to complement the lost details but introduces high-frequency noise at the same time. The ESRGAN output image has better image quality with more realistic and natural textures, and it tops the PIRM2018-SR challenge [23]. In the experimental section, we will cite the results of ESRGAN as a part of the comparison data.

The spatial resolution of FPDs is mainly determined by the pixel size, and a smaller pixel size leads to a higher spatial resolution. However, as a result of various factors such as process, cost, imaging quality, and imaging field of view, the pixel size cannot be reduced indefinitely at the actual product level. The common pixel size of amorphous silicon detectors is generally 100, 139, and 200 μm. The actual measured spatial resolution is worse than the theoretical spatial resolution corresponding to the pixel size. Types of scintillators (CSI, GdOS, etc.), scintillator thickness, and vapor deposition process will affect the actual spatial resolution. The measured spatial resolution of the detector with a nominal pixel size of 139 μm is typically between 150 to 190 μm.

We have developed a novel high-resolution X-ray FPD based on sub-pixel shift (SPS) technology, which can improve the inherent spatial resolution of the detector without degrading the image quality. We designed and implemented the underlying hardware and software for sub-pixel acquisition based on an amorphous silicon glass substrate, and users can easily control the sub-pixel acquisition accuracy of the detector through commands. Experiments show that the measured pixel size of this new high-resolution detector is reduced from 162 to 132 μm, that is, the basic spatial resolution of the detector is improved by 18.5%.

## 2. Evaluation Metrics

### 2.1. Basic Spatial Detector Resolution

"ISO-19232-5-2018, Non-destructive testing—Image quality of radiographs—Part 5: Determination of the image unsharpness and basic spatial resolution value using duplex wiretype image quality indicators" specifies a method of determining the total image unsharpness and basic spatial resolution of radiographs and radioscopic images [12]. The most important metric in this standard relevant to this work is $SR_b^{image}$ (basic spatial detector resolution), which is determined from the smallest number of the duplex wire pair.

The basic spatial detector resolution is measured with the duplex wire IQI (image quality indicator) directly placed on the detector without object.

The duplex wire IQI with up to 13 wire pairs can be used effectively with tube voltages up to 600 kV, and the specification of duplex wire-type IQI can be seen in Figure 1.

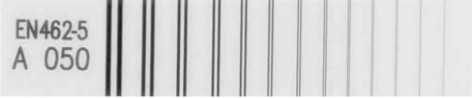

**Specification:**

- Duplex IQI consists of 13 or 15 wire pairs from 1D to 13D or 15D
- material for wires 1D to 3D is tungsten and for wires 4D to 15D is platinum
- distance between wires of each wire pair equals exactly diameter of wires
- wires are casted in a transparent, resistant and dimensional stable plastic
- standard and serial number are indelible casted and shown on each image
- design-type test for Duplex IQI is in process at BAM / Berlin

**Figure 1.** Specification of duplex wire-type IQI.

Measurement and calculation procedure for $SR_b^{detector}$: The duplex wire IQI should be placed directly on the detector with an angle between 2° and 5° to the rows/columns of the detector. No image processing shall be used other than gain/offset and bad pixel corrections.

If digital images are evaluated with a profile function, the element with the smallest wire number of the duplex wire pair, which is separable by a profile function with less than 20% modulation depth, is taken as the limit of discernibility for digital radiography. The profile function shall be evaluated from linearized pixel profiles. The measurement shall be done with the profile function of image-processing software across the middle area of the IQI image, integrating along the wires of about 30% to 60% of the duplex wires' length in order to obtain a robust repeatable value, but shall use a minimum of an 11-pixel width line profile to avoid variability along the length of the wires.

### 2.2. Image Quality Evaluation

"EN 12681-2:2017 Founding—Radiographic testing, part 2: Techniques with digital detectors" [24] specifies the recommended procedure for detector selection and radiographic practice, and the requirements for digital radiographic testing by either computed radiography (CR) or radiography with digital detector arrays (DDA) of castings. Three metrics are selected for the image quality evaluation of the proposed high-resolution X-ray flat panel detector: contrast sensitivity, $SR_b^{image}$ (basic spatial resolution of a digital image), and $SNR_n$ (normalized signal-to-noise ratio).

#### 2.2.1. Contrast Sensitivity

Unless otherwise agreed, the contrast sensitivity of digital images shall be verified by use of IQIs, in accordance with EN ISO 19232-1 or EN ISO 19232-2. A single wire IQI is shown in Figure 2.

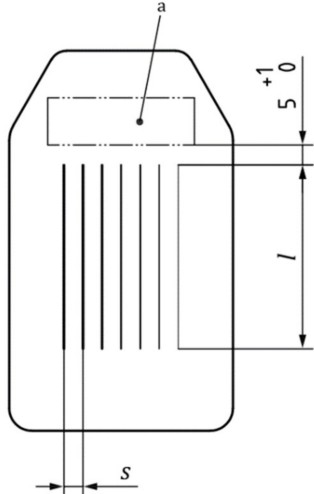

**Figure 2.** A schematic diagram of single wire IQI, in which, l: length of the wires, s: wire centerline spacing, a: space for identification marking.

The single wire IQI system is based on a series of 19 wires of different diameters, which are specified in Table 2 together with the relevant tolerances and the wire numbers. This series of wires has been subdivided into four overlapping ranges of seven consecutive wire numbers: W1 to W7, W6 to W12, W10 to W16, and W13 to W19. The seven wires in an IQI are arranged parallel to each other. The lengths of the wires (l) are 10, 25, or 50 mm. The single wire IQI shall be placed on the source side of the test object.

**Table 2.** Minimum image quality requirements for the visibility of wire IQIs for class A or B.

| Penetrated Thickness (mm) | | Minimum Wire IQI Value for Class A | Minimum Wire IQI Value for Class B |
|---|---|---|---|
| **Lower Thickness Limit** | **Upper Thickness Limit** | **IQI at Source Side** | **IQI at Source Side** |
| - | 1.2 | W18 | - |
| >1.2 for Class A and 0 for Class B | 2 | W17 | W18 |
| >2 | 3.5 | W16 | W17 |
| >3.5 | 5 | W15 | W16 |
| >5 | 7 | W14 | W15 |
| >7 | 10 | W13 | W14 |
| >10 | 15 | W12 | W13 |
| >15 | 25 | W11 | W12 |
| >25 | 32 | W10 | W11 |
| >32 | 40 | W9 | W10 |
| >40 | 55 | W8 | W9 |
| >55 | 85 | W7 | W8 |
| >85 | 150 | W6 | W7 |
| >150 | 200 | W5 | W6 |
| >200 | 250 | W4 | W5 |
| >250 | 380 | W3 | W4 |
| >380 | - | W2 | W3 |

### 2.2.2. Basic Spatial Resolution of a Digital Image

$SR_b^{image}$ (basic spatial resolution of image) corresponds to the effective pixel size and indicates the smallest geometrical detail that can be resolved in a digital image. For this measurement, the duplex wire IQI should be placed on the object (source side).

### 2.2.3. Normalized Signal-to-Noise Ratio

$SNR$ (signal-to-noise ratio) is the ratio of mean value of the linearized gray values to the standard deviation of the linearized gray values (noise) in a given region of interest

in a digital image. $SNR_N$ (normalized signal-to-noise ratio) is normalized by the basic spatial resolution $SR_b^{image}$ as measured directly in the digital image and/or calculated from measured $SNR_{measured}$.

$$SNR_N = SNR_{measured} \times \left(88.6\ \mu m / SR_b^{image}\right), \tag{1}$$

2.2.4. Minimum Image Quality Values

The radiographic techniques for film replacement are divided into two classes in EN 12681-2:2017:

– Class A: basic techniques
– Class B: improved techniques

Tables 2–4 show the minimum image quality requirements accordingly.

**Table 3.** Minimum image quality requirements for the visibility of duplex wire IQIs for class A or B.

| Penetrated Thickness (mm) | | Minimum DW Value for Class A IQI at Source Side | Minimum DW Value for Class B IQI at Source Side |
|---|---|---|---|
| Lower Thickness Limit | Upper Thickness Limit | | |
| - | 2 | D12 | D13+ |
| >2 | 5 | D10 | D13 |
| >5 | 10 | D9 | D12 |
| >10 | 24 | D8 | D11 |
| >24 | 40 | D7 | D10 |
| >40 | 55 | D7 | D9 |
| >55 | 85 | D6 | D9 |
| >85 | 150 | D6 | D8 |
| >150 | 200 | D5 | D8 |
| >200 | 250 | D5 | D7 |
| >250 | 380 | D4 | D7 |
| >380 | 150 | D4 | D6 |

**Table 4.** Minimum $SNR_N$ values for the digital radiography of aluminum, magnesium, and zinc.

| Radiation Source | Minimum $SNR_N$ for Class A | Minimum $SNR_N$ for Class B |
|---|---|---|
| $\leq 150$ kV | 70 | 120 |
| 150 to 250 kV | 70 | 100 |
| 250 to 500 kV | 70 | 100 |

## 3. High-Resolution Flat Panel Detector Design

### 3.1. The Proposed Image Detector Architecture

An indirect detector contains a layer of scintillating material that converts the X-rays into visible photons (light). Behind the scintillator, an array of photodiodes converts the light into an electrical signal. The array of photodiode pixels is similar in concept to a camera's image sensor: a high density of pixels creates a high-resolution image in which small features are clearly and sharply rendered. The stored charge of each pixel is proportional to the intensity of the incident X-rays. Under the action of the control circuit, the stored charge of each pixel is scanned and read out, and the digital signal is output after A/D conversion and transmitted to the computer for image processing to form a digital X-ray image. A based architecture demonstration of the indirect FPD is illustrated in Figure 3 [25].

The entire architecture of the proposed novel detector is shown in Figure 4, using CSI scintillator as the sensor; a-Si-based TFT as the converter; data operation part with integrated read, select, and pack; and a main state machine to control multiple modules and operation modes. Command passing through cmd process, buffer, and synchronization are designed. Udp/ip protocol stack and 1000 M wired network are used for all the command and data transfer.

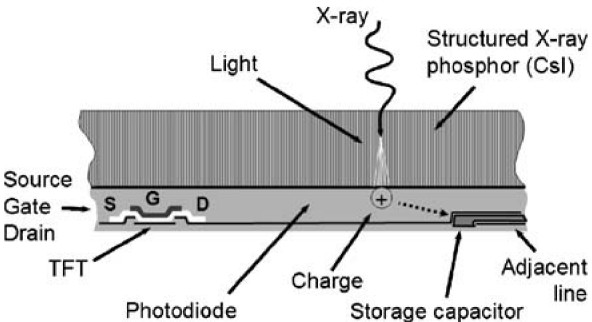

**Figure 3.** A based architecture demonstration of the indirect FPD.

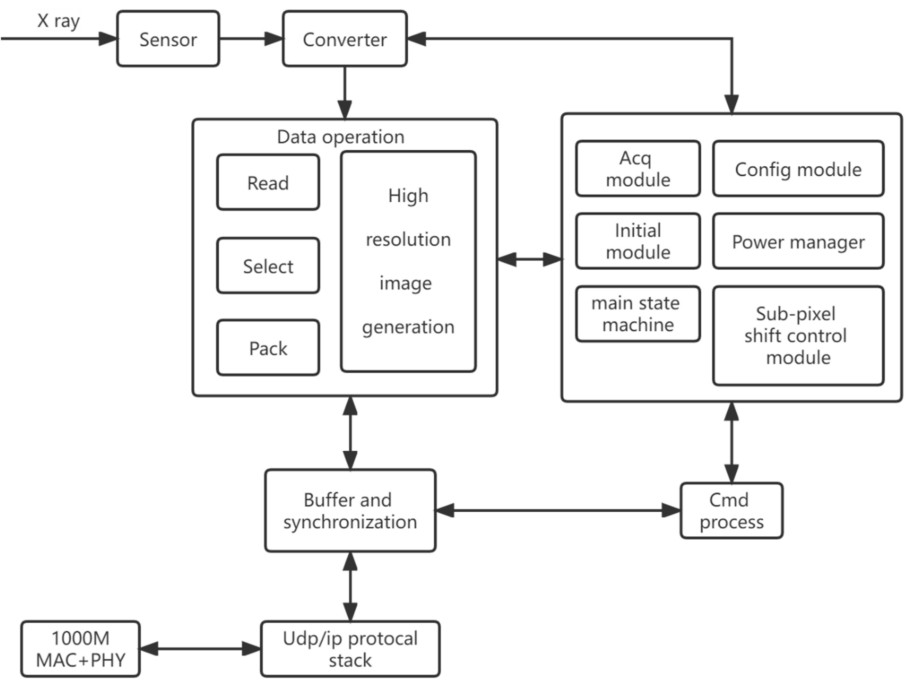

**Figure 4.** The entire architecture of our novel detector.

A circuit design diagram of some basic hardware modules of the detector is shown in Figure 5, which includes: (a) AD chip interface, (b) gate driver chip interface, (c) network interface, and (d) main control board.

*3.2. Sub-Pixel Shift Design*

Due to process and other constraints, the pixel size of the detector panel is limited. We use a sub-pixel shift design that allows each pixel to be spatially displaced in a controlled and precise sub-pixel scale in the XY direction, thus realizing pixel interpolation at the physical hardware level to improve the spatial resolution of the detector.

The procedure of sub-pixel shift is illustrated in Figure 6. It shows the four acquired original images and the high-resolution image composed.

In Figure 6, the resolution of the original image is S × S. Assuming that the inherent pixel size of the detector is L, then the distance between each two adjacent pixel centroids is L as well. Divide L equally into N parts to get the step size of each precise displacement. image a is the original image acquired at the original position (0, 0), image b is the original image acquired after moving L/2 distance to the right (X direction), image c is the original image acquired after moving L/2 distance to the bottom (Y direction), and image d is the original image acquired after moving L/2 distance to the right (X direction) and to the bottom (Y direction). Image e is the generated higher-resolution image with a resolution of 2S × 2S. In practice, when 2 × 2 mode is selected, first of all, the memory space inside the

detector is initialized with 4 times size of the original required, and all the pixel values are set to zero. Then, the pixel values of the four individual images are realigned to create a high-resolution image, as illustrated in Figure 6.

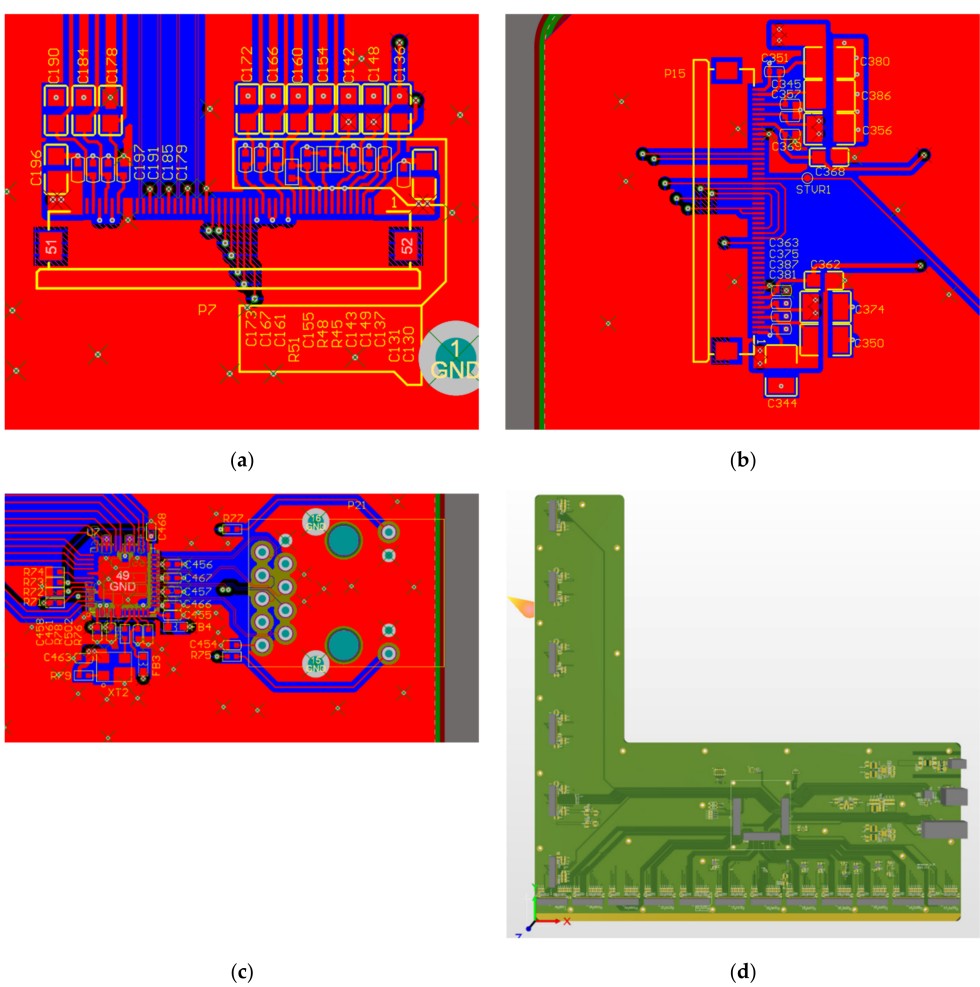

**Figure 5.** The circuit design diagram of some basic hardware modules of the detector: (**a**) AD chip interface, (**b**) gate driver chip interface, (**c**) network interface, and (**d**) main control board.

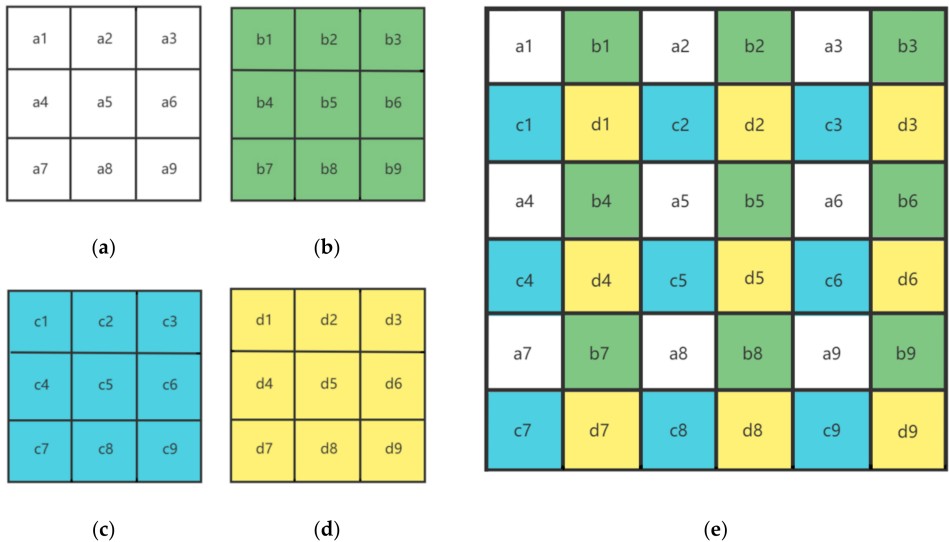

**Figure 6.** How high-resolution images are created by combining the pixels of individual native-resolution images. (**a**–**d**) are four acquired original images, and (**e**) is 2 × 2 mode higher-resolution image.

For modes other than $2 \times 2$, the operation is similar. We found in the actual test that the improvement of spatial resolution is limited after N > 4.

At the hardware level, we use an XY precision shift stage to fix the detector panel on it and complete the sub-pixel shift process. Image acquisition and image stitching are realized through integrated control and acquisition commands. The precision shift stage is shown in Figure 7. It uses noncontact optical encoders to measure the position directly at the platform with the greatest accuracy and achieves unidirectional repeatability to 0.05 μm and incremental linear encoder with 1 nm resolution. There are a variety of acquisition modes integrated in the main control board. In the application side, through the commands, the users can directly select different modes of precision for the high-resolution acquisition. Now, it supports a total of four modes: $1 \times 1$, $2 \times 2$, $3 \times 3$, and $4 \times 4$. The core parameters of the proposed sub-pixel shift FPD are shown in Table 5.

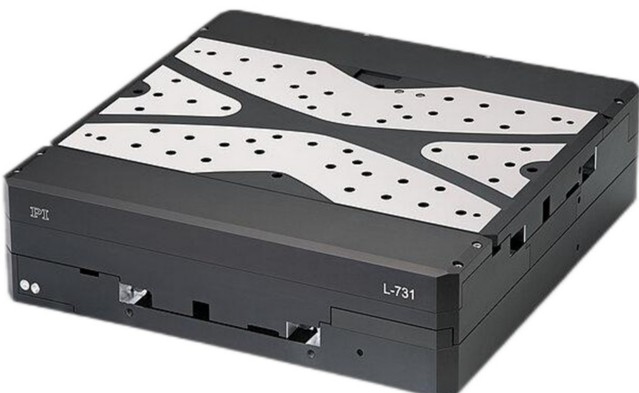

**Figure 7.** The XY precision shift stage.

**Table 5.** The core parameters of the proposed sub-pixel shift FPD.

| Parameter | Value |
| :---: | :---: |
| Type | a-Si |
| Scintillator | CSI |
| Number of rows | 3072 |
| Number of columns | 3072 |
| Pixel pitch (μm) | 139 |
| Imaging area (mm$^2$) | $430 \times 430$ |
| A/D bit | 16 |
| Frame per second (fps) | 6 ($1 \times 1$), 20 ($2 \times 2$) |
| Sub-pixel shift modes | $1 \times 1$, $2 \times 2$, $3 \times 3$, $4 \times 4$ |

## 4. Experiments

### 4.1. Experimental Environment

We have cooperated with Deepsea Precision Co., Ltd. [26], a manufacturer specializing in X-ray inspection equipment, to test the FPD and collect all the images from an actual X-ray environment. Figure 8 is a piece of industrial X-ray inspection equipment from Deepsea Precision Co., Ltd., with our FPD integrated into this testing system.

The basic configuration of the core imaging chain components from this typical X-ray inspection equipment of Figure 8 is shown in Table 6.

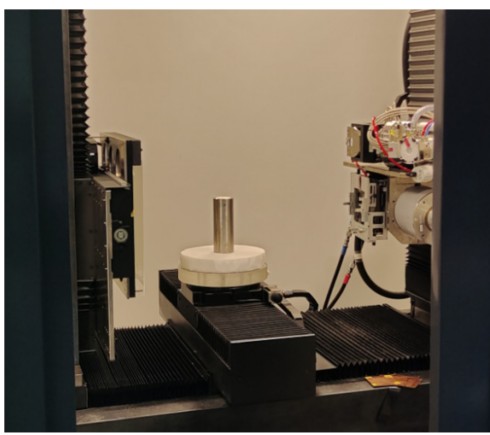

**Figure 8.** The industrial X-ray inspection equipment from Deepsea Precision Co., Ltd.

**Table 6.** The basic configuration of the core imaging chain components from the X-ray inspection equipment of Figure 8.

| Device Name | Brand/Model | Basic Configuration |
|---|---|---|
| X-ray emission device (macro-focus) | Gulmay/CF500 | 500 kV, focus size 0.4/1.0 mm |
| X-ray receiver device (flat panel detector) | Our sub-pixel shift FPD | 430 mm × 430 mm, pixel size 139 μm |
| Workstation software | Deepsea/DeepVISION | GPU-based architecture |

*4.2. Experimental Data*

4.2.1. Basic Spatial Detector Resolution Testing

The basic spatial detector resolution $SR_b^{detector}$ was tested according to ISO-19232-5-2018 under the following conditions: SDD (source detector distance) = 1000 mm, voltage = 90 kV, current = 1.3 mA. In our experiments, a software named DeepVision was used, and least squares curve fitting was adopted to calculate the SRb value in this software. The measuring procedure was carried out according to ISO-19232-5. We improved measurement precision by averaging the testing values of repeated measurements. Test results are shown in Table 7, and more detailed test images and data are shown in Figure 9. The results show that the basic spatial resolution of the novel detector proposed in this paper is greatly improved. The basic spatial detector resolution of the original image without any sub-pixel shift is 162 μm, the basic spatial detector resolution of the image after 2 × 2 sub-pixel shift is 140 μm, and the basic spatial detector resolution of the image after 4 × 4 sub-pixel shift is 132 μm. To summarize, the basic spatial detector resolution was improved by 13.6% after 2 × 2 shift and 18.5% after 4 × 4 sub-pixel shift.

**Table 7.** Test results of the basic spatial detector resolution $SR_b^{detector}$.

| The Basic Spatial Resolution of the Detector | Original Image | 2 × 2 Sub-Pixel Shift Image | 4 × 4 Sub-Pixel Shift Image |
|---|---|---|---|
| $SR_b^{detector}$ | 162 μm (D8) | 140 μm (D9) | 132 μm (D9) |

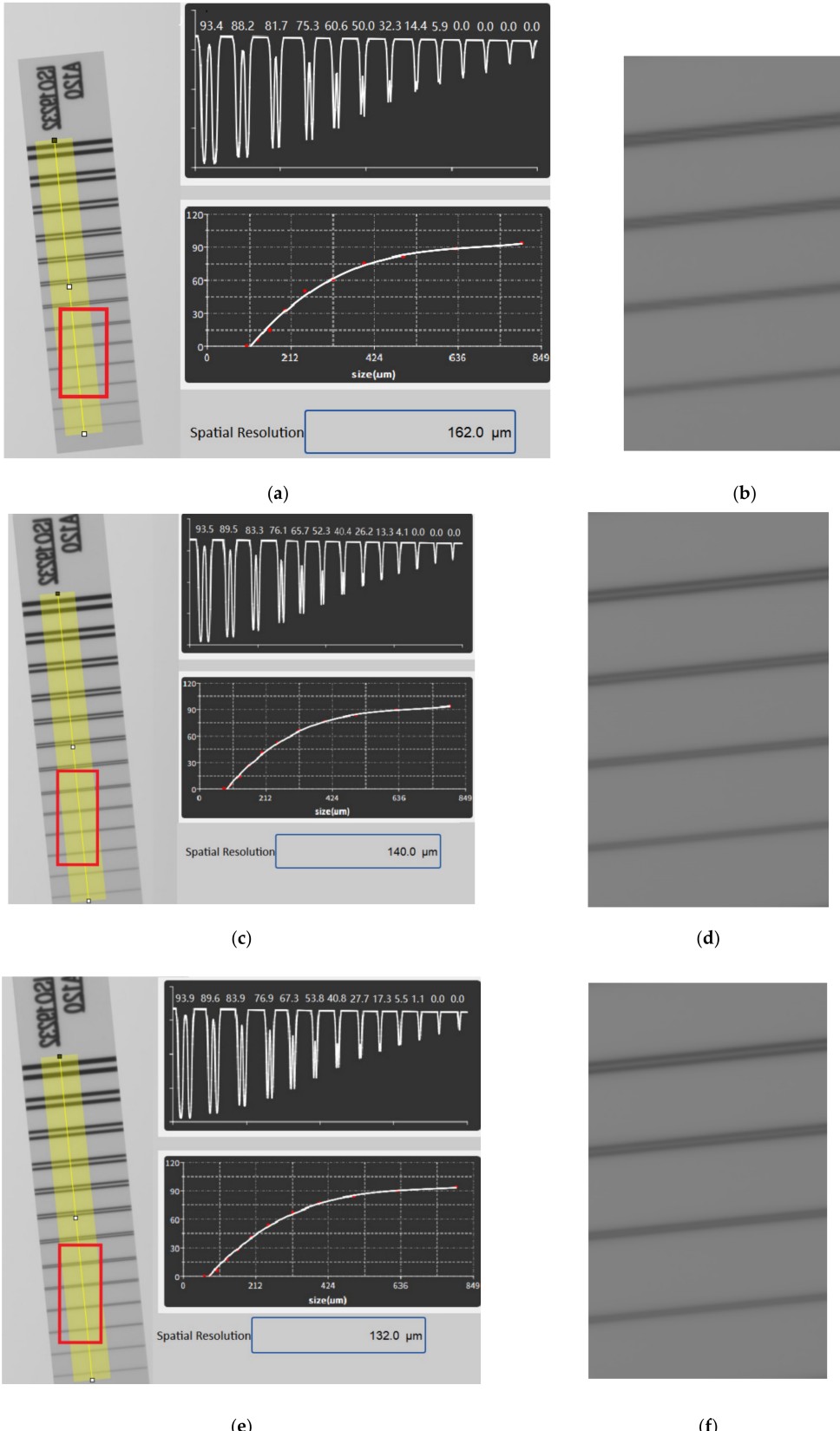

**Figure 9.** The basic spatial detector resolution $SR_b^{detector}$: (**a**) original image, (**c**) $2 \times 2$ sub-pixel shift image, (**e**) $4 \times 4$ sub-pixel shift image; (**b,d,f**) are the corresponding local enlargement of the boxed areas of the duplex wire IQI.

### 4.2.2. Image Quality Testing

According to the standard EN 12681-2:2017, the duplex wire IQI and single wire IQI are fixed on a step wedge for image quality testing. The step wedge is aluminum, and the thinnest layer is 10, then 20, 40 mm, and so on. The testing was conducted at 20 mm thickness. The test condition is: SDD (source detector distance) = 1000 mm, voltage = 140 kV, current = 1.5 mA. A total of three tests are conducted: single wire IQI, duplex wire IQI, and $SNR/SNR_n$.

Test 1: Single Wire IQI Result

The test results are shown in Figure 10. The visibility levels of the single wire IQI on the original image and the resolution-enhanced image after sub-pixel shift are the same, both being W14. This result indicates that the proposed sub-pixel-shift-based method does not cause a decrease in contrast sensitivity.

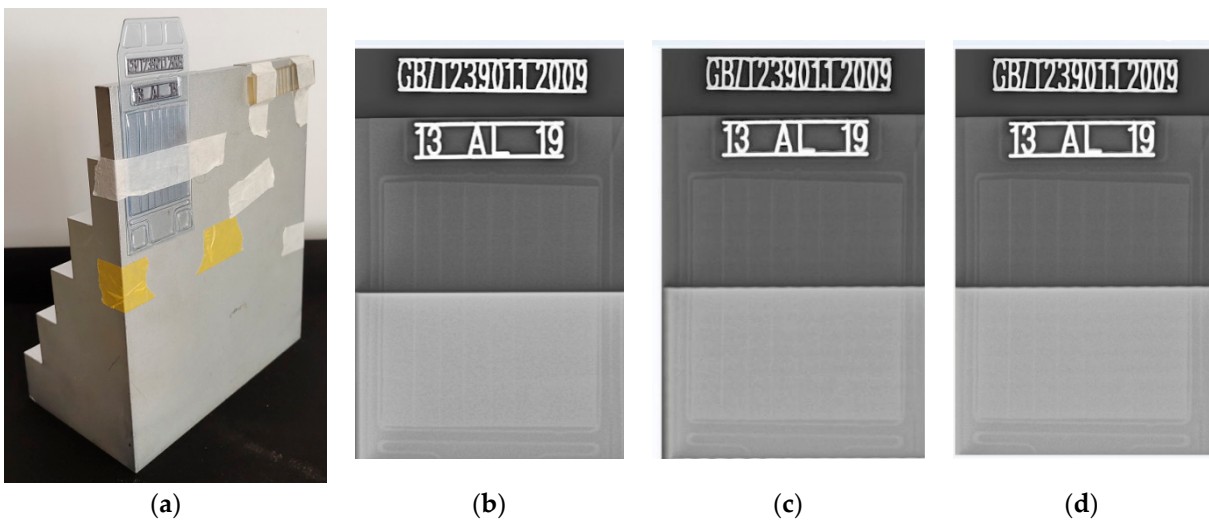

| (a) | (b) | (c) | (d) |

**Figure 10.** Test results of the contrast sensitivity: (**a**) experimental device, (**b**) original image, (**c**) 2 × 2 Sub-Pixel Shift Image, (**d**) 4 × 4 Sub-Pixel Shift Image.

Test 2: Duplex Wire IQI Result

For the image basic spatial resolution test, the duplex wire IQI should be attached to the step wedge with a certain magnification to evaluate the spatial resolution performance when imaging the actual object. Test results are shown in Table 8, and more detailed test images and data are shown in Figure 11.

**Table 8.** Test results of the image spatial resolution $SR_b^{image}$ of the detector.

| The Basic Spatial Resolution of the Detector | Original Image | 2 × 2 Sub-Pixel Shift Image | 4 × 4 Sub-Pixel Shift Image |
|---|---|---|---|
| $SR_b^{image}$ | 159 μm (D8) | 137 μm (D9) | 132 μm (D9) |

The results show that the image spatial resolution of the novel detector proposed in this paper is greatly improved. The image spatial resolution of the original image without any sub-pixel shift is 159 μm (D8), which does not meet the minimum image requirement according to Table 3. In contrary, the image spatial resolution of the image after 2 × 2 sub-pixel shift is 137 μm, and the image spatial resolution of the image after 4 × 4 sub-pixel shift is 132 μm. That is to say, the image quality achieved by the proposed sub-pixel shift method has been improved to meet the requirements for Class A.

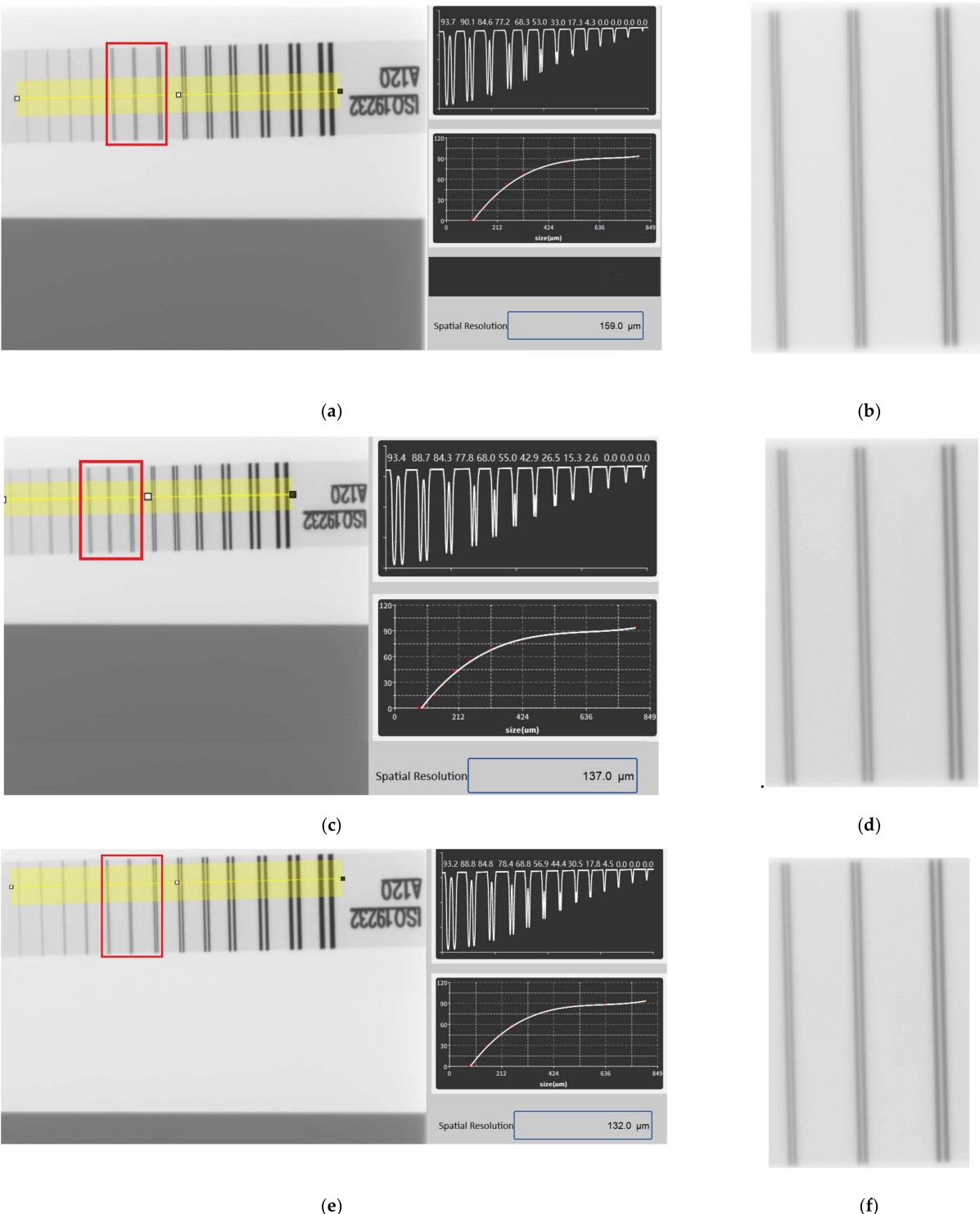

**Figure 11.** The image spatial resolution $SR_b^{image}$ of the detector: (**a**) original image, (**c**) $2 \times 2$ sub-pixel shift image, (**e**) $4 \times 4$ sub-pixel shift image; (**b**,**d**,**f**) are the corresponding local enlargement images of the boxed areas of the duplex wire IQI.

Test 3: $SNR/SNR_n$ Result

According to the standard, $SNR/SNR_n$ is calculated in different grayscale distribution areas, and an $SNR/SNR_n$ graph is generated, as shown in Figure 12 with the average grayscale value of the selected area in the horizontal coordinate and the $SNR/SNR_n$ value in the vertical coordinate. It can be concluded that the proposed sub-pixel-shift-based

high-resolution FPD in this paper improves slightly in $SNR$ and significantly in $SNR_n$ due to the great improvement in spatial resolution.

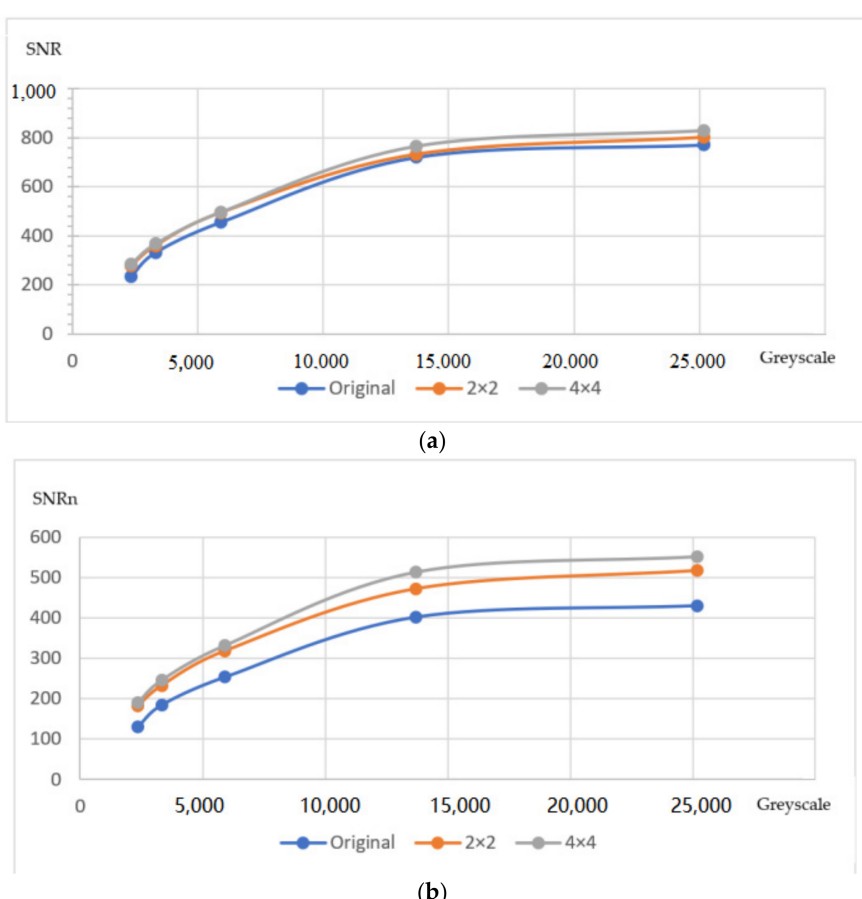

**Figure 12.** The $SNR/SNR_n$ of the detector: (**a**) $SNR$, (**b**) $SNR_n$.

### 4.2.3. Comparison with Software Interpolation Methods

Some of the software-based super-resolution methods have been discussed previously, among which the ESRGAN algorithm is currently considered to be the most effective method. We compared the results of the ESRGAN algorithm with ours, as shown in Figure 13. ESRGAN improves the overall spatial resolution, but introduces local artifacts and noise, which can seriously affect the inspection results and is therefore not applicable. The arrows indicate artifacts produced by ESRGAN algorithm. In the proposed method, an XY precision shift stage is adopted to move the detector. The FPD frame rate is 6 FPS, which means 6 frames per second. The shift stage moves the detector to the next position during each image readout time. In other words, the movement of detector does not occupy extra time. In general, for $2 \times 2$ mode, the total time for generating a high-resolution image is less than 1 s. For $4 \times 4$ mode, the total time is about 3 s. For ESRGAN method, the total time for upsampling a $3072 \times 3072$ image to $6144 \times 6144$ is about 800 ms in our experiments.

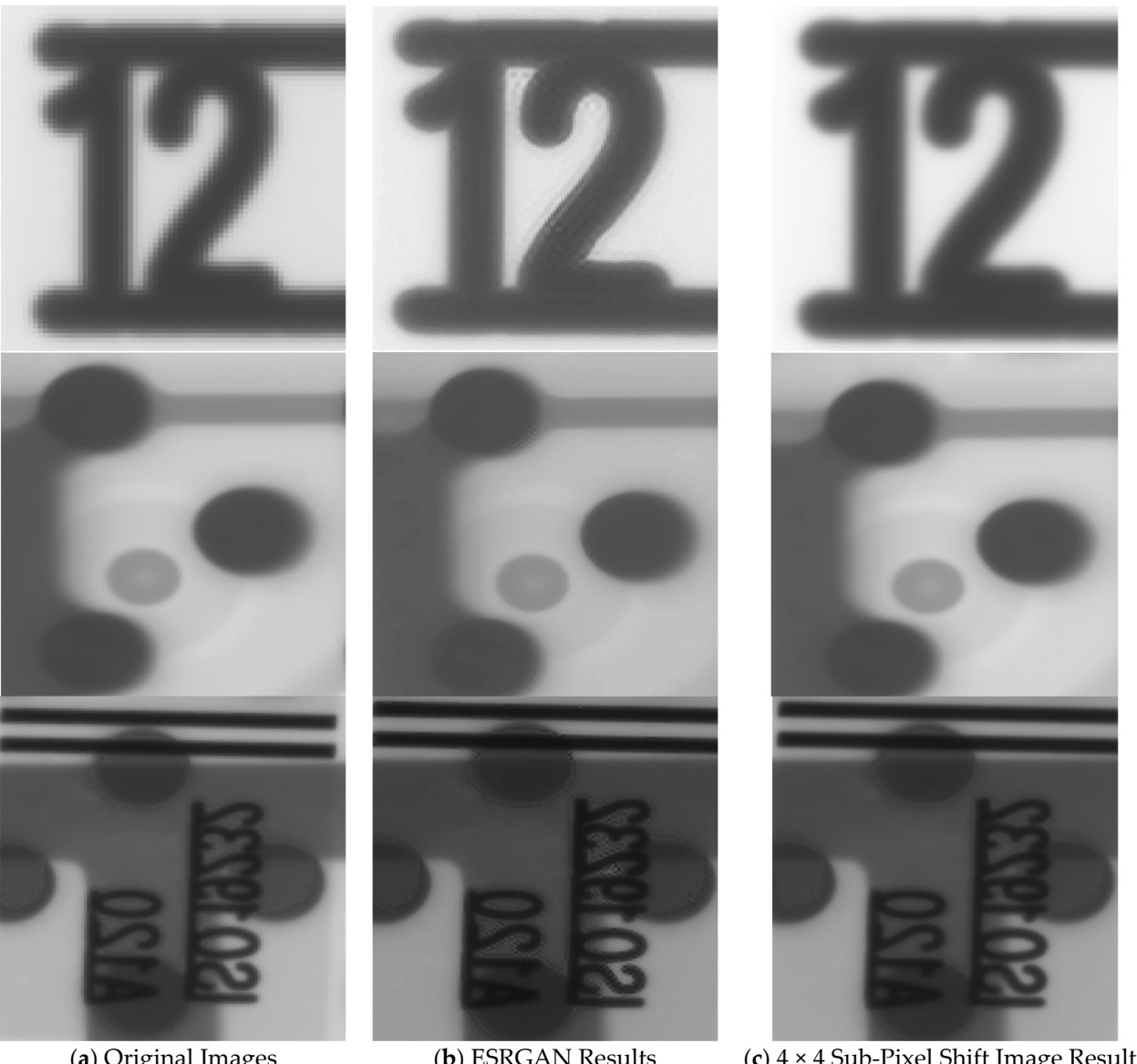

(**a**) Original Images        (**b**) ESRGAN Results        (**c**) 4 × 4 Sub-Pixel Shift Image Results

**Figure 13.** The results of the ESRGAN algorithm and ours on real object images. ESRGAN improves the overall spatial resolution but introduces local artifacts and noise. The arrows indicate artifacts produced by ESRGAN algorithm.

## 5. Conclusions

In this paper, we describe a novel sub-pixel shift (SPS)-based X-ray flat panel detector (FPD) that can achieve high resolution while maintaining high SNR. In comparison with the existing techniques, the proposed high-resolution FPD has the following features: (i) a sub-pixel-shift-based acquisition and data composition strategy are integrated in the detector; (ii) the pixel size of the detector was reduced from 162 to 132 μm, that is, the basic spatial detector resolution was improved by 13.6% in the simplest 2 × 2 sub-pixel shift mode, and by 18.5% in 4 × 4 sub-pixel shift mode; (iii) for X-ray images, the test results show a great improvement in image spatial resolution, as well as high $SNR_n$.

The main goal of X-ray image processing is to avoid introducing artifacts. This is especially important for X-ray non-destructive testing. Image super-resolution by software algorithms, including interpolation methods, reconstruction-based methods, and deep learning methods, have been proposed and studied for many years. Some of them can achieve great performance on visible light images. However, they have poor performance on X-ray images with high dynamic range and internal noise. Some may cause severe image

distortion and artifacts, while others may enlarge the image noise. The poor performance of these high-resolution methods might lead to false alarms for X-ray non-destructive testing. The proposed method has been applied to the testing system, and the performance has been demonstrated by experiments. The results show that our method is effective. Both spatial resolution and SNRn have been improved without introducing artifacts.

In the future, we plan to apply the methodology in this work to other types of X-ray detectors. In addition, we will try to increase the frame rate of the detector using a 10 Gigabit network or fiber optic interface, which allows the sub-pixel acquisition process to be done faster.

**Author Contributions:** Conceptualization, J.L. and J.H.K.; methodology, J.L.; software, J.L.; validation, J.L.; formal analysis, J.L.; investigation, J.L. and J.H.K.; resources, J.L.; data curation, J.L.; writing—original draft preparation, J.L.; writing—review and editing, J.H.K.; visualization, J.L.; supervision, J.H.K.; project administration, J.L.; funding acquisition, J.L. All authors have read and agreed to the published version of the manuscript.

**Funding:** This research was supported by Guangdong Science and Technology Project, Grant Nos. 2020A0505100012, 2017B020210005, and BK21FOUR; and Creative Human Resource Education and Research Programs for ICT Convergence in the 4th Industrial Revolution.

**Institutional Review Board Statement:** Not applicable.

**Informed Consent Statement:** Not applicable.

**Data Availability Statement:** Not applicable.

**Acknowledgments:** Thanks to Deepsea Precision Co., Ltd. and for their X-ray inspection equipment and support for the FPD testing. Thanks to all the people who contributed to the FPD and image testing. Special thanks to Stan, a colleague of J.L., who did a lot of work in building the testing environment.

**Conflicts of Interest:** The authors declare no conflict of interest.

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
