# Peer review of "A Novel Sub-Pixel-Shift-Based High-Resolution X-ray Flat Panel Detector"

_coatings, doi:10.3390/coatings12070921_

Round 1

Reviewer 1 Report

A well-written manuscript, with a sound panel detector design,
minimizing software processing of image acquisitions by addressing
all sub-pixel shift issues by hardware-based solutions.

A suggestion (maybe only requiring some further, minimal discussion):

The precision of the sub-pixel shift task highly depends on the exact alignment of sub-pixel acquisitions (it seems to be assumed as
perfect). An estimation of the resulting error on the SNR, coming
from the hardware and setup imprecisions could be more detailed,
making a plot showing the reduction in the SNR for a fixed increase in resolution with varying precision conditions from the hardware and acquisition protocol. You could show that it complies with some
maximum tolerance in specific applications.

Please address the following corrections/comments:

in ABSTRACT, line 17, you wrote "2X2", but it is clearly "4x4"
as stated in line 315:  4x4 sub-pixel shift for pixel size reduced
to 132 microns.

Line 142: you wrote "nomina pixel", it should be "nominal pixel"

Line 161   Whe first writing "IQI", expand its definition, that is
write   "IQI (Image Quality Indicator)" --- or Index?

Line 262   "slightly in SNRand"     Write:  "slightly in SNR and"

Lines 306-309  
Fix any pending issue or change to black this paragraph written in red
If you want to stress the information, you may use a bold font, but verify
if it is allowed by the editors.

Lines 371-378   Same comments.

394   "The main principal of X ray image processing is to avoid introducing artifacts".
      Write instead:   "The main goal of X ray image processing is to avoid introducing artifacts".

I recommend acceptance after addressing he above corrections and
comments.

Reviewer 2 Report

In general the paper is well written and interesting. And it has enough merits to be considered for publication.

I have only some minor comments.

-          I am not sure on how useful are Figure 1 and Table 1 to the readers. While it is somewhat interesting information about the field, I find it distracting and causing the paper to lose its focus on the actual results and contribution of the authors.

-          Figure 2, does not help much to a reader to visualize what or how the FPDs work While it clearly shows how the two main types work, its also something that may not really require a full figure. But can easily explained in text. It also could be replaced with a more attractive figure which also illustrated the actual aspect of the systems. In other words, I think the figure is too big and provides not enough information for its size on the paper.

-          Also related to the previous points. I find in general, that that the text can be shortened significantly in several places, without sacrificing useful information, but gaining in clarity, conciseness.

-          I find the way of presenting the metrics in 2.2.2 and 2.2.3 a bit confusing.

-          I would like a more systematic comparison with software methods. Software super resolution research works usually use several metrics to evaluate and compare different techniques. Maybe some of these comparison methods can be used here.

Author Response

This manuscript is a resubmission of an earlier submission. The following is a list of the peer review reports and author responses from that submission.

Round 1

Reviewer 1 Report

This paper describes the work the authors have undertaken to incorporate dithering to create super-resolution radiographs using a standard flat panel detector. The super-resolution images are created by a very simple interlacing technique which is implemented in hardware, and improvements in resolution  of up to 18% were demonstrated. While the paper is generally quite thorough, I cannot recommend this for publication since the interlacing technique is simple and well known, and as such the paper will not be of significant interest to the readers of Sensors.

Specific points:

1. Why did the authors choose a simple interlacing method? There are alternative methods commonly used in astronomy that perform better than simple interpolation (as well as avoiding artefacts seen in the CNN methods). For example Drizzle (Fruchter et al 2002 PASP 114 144) is widely used.

2. The authors suggest that SNRn is improved due to an improvement in resolution. However, for 4x4 super-sampling, you are taking extra 16 frames. SNRn could therefore be improved by a simple averaging of those frame to reduce noise (without improving resolution). The authors should compare how SNRn improved for simple averaging

3. What is the time overhead for moving the detector? Is this minimised in the hardware implementation compared with a software implementation?

4. Figure 6 is not particularly useful. This should be replaced with a flow diagram outlining the procedure to generate a super-resolution radiograph

5. Describe in more detail how SRb was measured. Which software was used and what interpolation function was used? What is the uncertainty (standard error estimate) on the measured SRb?

6. The authors mention that high DQE is maintained. However no measurements of DQE are presented. Please state the measurements of DQE

7. Figure 7 showing the circuit board layouts should be removed.

8. Ensure all acronyms are defined including DR and SMT

9. Figure 1 shows the relative performance of different x-ray detector technologies. How is performance calculated? Is it a combination of DQE and resolution for example?

8. The images in Figure 11 and Table 8 are not clear. Please improve the contrast so that the features can be clearly seen, and ensure the same contrast is used for images of the same target. Also local enlargements are displayed, without showing where in the full images these enlargements were taken

9. Figure 13: the axes should have titles and units

Reviewer 2 Report

The manuscript has presented an x-ray flat panel detector method based on the sub-pixel shift approach. The manuscript is well documented and the results are promising. I would like to recommend the article in its present form. However, the authors must provide full names for all abbreviations including at Abstract for better understanding of the readers.

Reviewer 3 Report

Authors present a method dedicated to the improvement of Spatial Resolution in X-ray Flat Panel Detector based on sub-pixel shift mechanism, proving that this technique provide an increment of 13-18 % with respect to the standard method.

The manuscript is clearly written and the conclusions are supported by results.

Please correct the misprintings on lines 106 (dot, not comma) and 232 (CSI, not Csi)  and add both the name and units on the x-y axis of Figures 13 (a,b).

Overall the manuscript can be considered for publication on Sensors, however, being the ESRGAN and Sub-Pixel methods similar, I suggest the authors to provide also a comparison both in terms of overall time required to obtain the image and data computational requirements to implement the paper.